# Long Term Findings Concerning the Mental and Physical Condition, Quality of Life and Sexuality after Laparoscopically Assisted Creation of a Neovagina (Modified Vecchietti Technique) in Young MRKHS (Mayer-Rokitansky-Küster-Hauser-Syndrome) Patients

**DOI:** 10.3390/jcm10061269

**Published:** 2021-03-18

**Authors:** Katharina Rall, Bernadette Schenk, Norbert Schäffeler, Dorit Schöller, Andrina Kölle, Birgitt Schönfisch, Sara Y. Brucker

**Affiliations:** 1Department of Women’s Health, Women’s University Hospital, Calwerstr. 7, 72076 Tübingen, Germany; Bernadette.schenk@web.de (B.S.); dorit.schoeller@med.uni-tuebingen.de (D.S.); Andrina.koelle@med.uni-tuebingen.de (A.K.); Birgitt.schoenfisch@uni-tuebingen.de (B.S.); sara.brucker@med.uni-tuebingen.de (S.Y.B.); 2Division of Psychosomatic Medicine and Psychotherapy, Department of Internal Medicine, Tübingen University Hospital, Osianderstr. 5, 72076 Tübingen, Germany; norbert.schaeffeler@med.uni-tuebingen.de

**Keywords:** MRKHS, mental health, physical health, sexuality, quality of life, depression, infertility

## Abstract

The Mayer-Rokitansky-Küster-Hauser-syndrome (MRKHS) is characterized by a congenital uterine and vaginal aplasia. A large body of literature reports that a diagnosis of MRKHS has a variety of psychological effects on patients and doubts about female identity. The aim of the underlying study was to detect the patient-reported physical and mental health and sexual function before and after laparoscopically assisted creation of a neovagina. 160 women with MRKHS who underwent this type of surgery between September 2009 and December 2015 were invited to complete the questionnaires. Packages consisting of six questionnaires were handed out before surgery, six and 12 months after surgery. Data from 82 patients could be included in the study. Patients had a mean age of 19.9 years at inclusion in the study. We detected an impairment of the health-related mental quality of life. There was no higher risk for psychological disorders. MRKHS patients show similar self-acceptance and normal body image compared to the general population. The sexual function is limited before surgery and normalizes after surgery. Useful factors for coping with the disease are an interdisciplinary approach in diagnostics and treatment, psychosocial adaptation as well as a supportive social environment.

## 1. Introduction

The Mayer-Rokitansky-Küster-Hauser-syndrome (MRKHS) is characterized by the congenital absence of a functioning uterus and the upper 2/3 of the vagina [1]. With an incidence of 1:4.000 to 1:6.000 female livebirths, the MRKHS belongs to the group of rare (orphan) diseases [2,3,4]. The MRKHS may occur isolated (type I MRKHS), but associated malformations mainly of the kidneys and lower urinary tract and the skeletal system have been described among others (type II MRKHS). In a cohort of 346 MRKHS patients we identified type 1 MRKHS in 53.2%, type 2 MRKHS in 41.3% and MURCS (Mullerian-renal-cervicothoracic-somite abnormalities) in 5.5% [5]. 

The MRKHS has increasingly become targeted by medical research in recent years due to particular attention to rare diseases and emerging therapeutic options such as uterine transplantation [6].

The diagnosis of vaginal and uterine aplasia is usually not made before puberty, because the ovarian function and external genitalia are normal. Primary amenorrhea with or without coital problems is the main reason for consulting a gynecologist. The diagnosis therefore often falls into a phase of life that is characterized by both physical and psychosocial changes and crises, and essentially shaped by the development of female identity and sexuality [6,7,8]. In this period of life with the transition from childhood to adulthood, the diagnosis of MRKHS presents a great emotional and mental burden for patients, in particular due to the infertility and the lack of cohabitation ability [6]. Due to vaginal aplasia sexual intercourse is not possible without any type of conservative or surgical procedure for the creation of a neovagina, and the missing uterus prevents the patients from carrying a pregnancy. Accordingly, the increase of quality of life, including a possibility of cohabitation [9] is among the main objectives of the therapy [6]. The majority of patients presents with the wish to become sexually active and decides to either perform non-surgical self-dilation and/or intercourse, or to undergo surgery. To date, the non-surgical dilation with vaginal dilators according to Frank [10] is the first choice of treatment [11,12]. Dilation can also be performed surgically with Vecchietti’s procedure [13] and its modifications [14,15] and involves a laparoscopic attachment of a traction device to the abdomen with subperitoneal sutures and a plastic ”olive” placed on the vaginal dimple [11]. Our modified technique creates a neovagina of adequate size and secretory capacity for normal coitus, requiring no prolonged dilation postoperatively, even in the absence of sexual intercourse. The procedure is fast, effective and minimally traumatic, has a very low long-term complication rate and provides very satisfactory long-term functional results [14]. 

Recently, it could be shown, that in comparison to controls, women with MRKHS with a non-surgically or surgically created neovagina did not differ in psychological and relational functioning but reported lower sexual esteem and more negative genital self-image, intercourse-related pain, clinically relevant sexual distress and sexual dysfunction [11].

A large body of literature reports that a diagnosis of MRKHS has a variety of psychological effects on patients, including depression, self-esteem and body image disturbances, feelings of incompleteness, and doubts about female identity [7,16,17,18,19,20].

In a Chinese study 24.1% of the MRKHS patients experienced moderate to severe anxiety symptoms compared to healthy controls. Patients with MRKHS manifested more severe anxiety symptoms than healthy women. Negative self-evaluation of femininity, neurotic personality traits and coexisting depressive symptoms were more prevalent in anxious patients [16].

The bodily characteristics of these syndromes can leave individuals questioning their femininity, attractiveness, and worth as a woman and sexual partner [8,20,21]. Fliegner et al. were able to show that somatic and psychological factors are crucial to sexual well-being. Professionals should provide an interdisciplinary approach depending on patients’ individual needs in order to promote high levels of sexual well-being [20].

Bean et al. described four critical points of time for patients with MRKHS: point of time when diagnosis is made, at non-surgical/surgical therapy, first relationship/sexuality and wish to become a mother [7]. 

The aim of the underlying study was to detect the patient-reported mental burden, physical being as well as the ability to be in a relationship and sexually active, before and after laparoscopically assisted creation of a neovagina, using a self-developed as well as standardized questionnaires. 

A special focus was put on the effect of the surgical treatment as well as on the factor time concerning coping strategies for individuals with MRKHS. Screening questionnaires were used in order to identify the rate of associated psychiatric diseases.

Predictive signs should be identified in order to be able to offer psychological treatment and support depending on the individual needs.

## 2. Study Group and Methods

### 2.1. Design and Participants

All women with MRKHS type 1 or 2 presenting to the center for rare female genital malformations within the Department of Women´s Health at the University Hospital in Tübingen between September 2009 and December 2015, who underwent a laparoscopically assisted creation of a neovagina as previously published [15] were invited to participate in the study by completing the questionnaires. Further inclusion criteria were sufficient knowledge of the German language and informed consent. After informed consent the patients were given the set of questionnaires according to the timepoint: 1. before surgery, preoperative version, timepoint 1 (TP1) and during the follow-up visits 6 months (timepoint 2 (TP2)) and 12 months (timepoint 3 (TP3)) after surgery. 

A gynecologic examination including inspection of the external genitalia and palpation of the vaginal dimple preoperatively as well as measurement of the anatomic (without pressure) and functional (with maximal digital pressure) length and width after surgery was routinely performed. We routinely performed PAP smears, microbiological and HPV smears at TP2 and 3 and evaluated the grade of epithelialization.

The packages for all three timepoints consisted of six questionnaires: the self-developed NeoCope, the German version of the Patient Health Questionnaire (PHQ-D), the questionnaire concerning body image (FKB-20), the German version of the Female Sexual Function Index (FSFI-d), the short version of the SF-36 Health Survey (SF-12) and the Scale for the detection of self- acceptance (SESA). Except for the NeoCope, which has been attached at the end of the manuscript, all other questionnaires are accessible online.

The evaluation was performed as part of the unpublished medical thesis work of B.S. B.S. fully agreed to include the present data.

### 2.2. Measures

#### 2.2.1. Neocope

This extensive paper-and-pencil questionnaire was designed 2004 by the Department of Women’s Health and the Department for Psychosomatic Medicine to cover demographic and clinical characteristics of the study participants such as age, duration of the partnership and educational level, details about the diagnostic procedures and others and included standardized as well as self-developed scales and open questions for individual comments. The name is self-created by using parts of the words “coping” and “neovagina”.

There is a preoperative short version including 24 items as well as the versions 6 and 12 months after surgery including 47 and 38 items respectively (see Appendix A).

#### 2.2.2. FKB-20—Body Image

1996 the psychologists Clement and Löwe from Heidelberg developed the body image questionnaire (FKB-20) for the diagnosis of body image disturbances and for the identification of subjective aspects of body perception [22]. The questionnaire consists of 20 five-staged items and includes topics like movement, vitality, attractivity and subjective coherence. Body image is divided into two scales, each including 10 items: The scale Rejecting body image (RBI) detects the evaluation of the external body image and rates the well-being within the own body evaluations of an individual’s body image regarding appearance and well-being. It is known to be relatively sensitive in detecting body centred disturbances. 

The scale vital body dynamics (VBD) focuses on the movement-related aspect of the body image (strength, fitness, health, sexuality). 

A high value on RBI correlates with a negative body image, whereas a high value on VBD correlates with a powerful and positive body image.

The internal consistency by Cronbachs α is given as high for the RBI scale with 0.84 and for the VBD scale with 0.79 [23]. 

The questionnaire was chosen with the aim to detect changes in body image before and after surgery. 

#### 2.2.3. German Version of the Female Sexual Function Index (FSFI-d)

The American Psychiatrist Prof. Dr. Rosen published in 2000 the Female Sexual Function Index (FSFI), which was translated into German in 2003 at the university of Freiburg and validated for the German region including a large sample size (*n* = 1243) [24,25]. After a process of item reduction, 19 items with optimal psychometric properties, sampled from each of six domains (desire, arousal, lubrication, orgasm, satisfaction and pain), were selected. A scoring system was developed for obtaining individual domain scores and a full scale score [24]. Higher scores mean better functionality. 

The internal consistency of the subscales according to Cronbachs α is given with acceptable to excellent values of 0.75 to 0.95 [25].

The FSFI-d questionnaire was selected in order to compare pre- and postoperative sexual function and its limitations, although it focuses on the four weeks preceding the completion.

### 2.3. Short Form of the Health Survey (SF-12)

The SF-12 is a short form of SF-36, translated and psychometrically validated for the German speaking regions and is a standardized tool for the acquisition of the subjective quality of life [26]. It tests the health-related quality of life only on a physical (PSC) and a mental (MCS) scale. The completion of the items differentiates, it may be binary with “yes and no” or on scales with up to 6 possible answers. The data of the MRKHS patients are compared to 4 reference sample groups, two healthy ones and two with acute or chronic diseases, each stratified by age (14–20 and 21–30 years of age). The reference sample groups include females and males.

The SF-36 has been shown to have good psychometric properties [26]. 

The short form was used in order to evaluate the subjective quality of life before and after laparoscopically assisted creation of a neovagina. 

### 2.4. Self-Acceptance Registration Scale (SESA)

It is used for the evaluation of the self-acceptance of a patient and is suitable in order to detect changes during the course of therapy. The value can be assigned to three categories related to the mean of the sum score of the non-clinical reference sample: with a sum score >1 SD, no differentiated examination is necessary (scores between 97 and 145). With a test value between −1 SD and −2 SD, a differentiated examination should be performed (scores between 81 and 96); a test value < −2 SD indicates a depressive disorder (scores between 29 and 80).

The internal consistency according to the Kuder-Richardson formula 20 is defined as r_tt_ = 0.83 [27]. With the SESA questionnaire we wanted to evaluate if there is a reduced self-esteem or self-acceptance before and after surgery. 

Ethics approval and consent to participate: Ethical approval has been given by the University Hospital Tübingen Ethic’s Commission (project number 554/2013BO2). Prior to a subject’s participation in the study, a written informed consent form was sought and gained from all participants of the study. All patients were given an information sheet together with the informed consent with the advice that they could revoke their consent at any time without giving any reasons. The signed consent forms of all participating patients and separately their pseudonymized questionnaires are kept safe in the Department of Women’s Health Study Coordination Office. In case of minors the agreement and written informed consent of at least one guardian was needed.

Licences of the publisher (Hogrefe-Verlag) were bought for the FKB-20, SF-12 and SESA questionnaire.

### 2.5. Statistical Analysis

Data input and evaluation were done with the IBM SPSS statistics 26 software (IBM Corp., Armonk, NY, USA). In order to reproduce the distribution of continuous variables, number (*N =* whole random sample, *n =* partial sample), arithmetic means (*M*), standard deviation (*SD*), median (*Mdn*), minimum (*Min*) and maximum (*Max*) are used. For dichotomous data sets and questions with choices, also with possible multiple selections, numbers and ratios were calculated. 

Normal distribution was checked with the Shapiro-Wilk test (data not shown), because it has a good power also in cases of small sample sizes (*n* < 50). *p* < 0.05 indicates that we refuse the null hypothesis and no normal distribution can be assumed, while *p* > 0.05 indicates normal distribution. Values of the standardized questionnaires at the single points of measurement were tested for significance with cut-off-values or reference samples in case of normal distribution with the one-sample *t*-test (= Student‘s *t*-Test). The effect size according to Cohen’s was used. 

In cases of missing normal distribution, the Wilcoxon signed-rank test was performed. Thereby, the effect size rank-biserial correlation (*r*_bis_) was interpreted according to Cohen. In order to evaluate changes over the time, the two-factor analysis of variance for ranks according to Friedman for related samples was used. The effect size Kendall’s *W* was interpreted for the Friedman-test according to the literature. In addition, the Conover’s Post-hoc-Test was calculated in order to examine which timepoints of measurement differ from each other. 

The Chi-square contingency analysis (χ^2^) was used once, in order to check if two categorial data sets are related.

## 3. Results

### 3.1. Patients

160 patients were asked to participate in the study at first presentation to our centre. 

Three groups were built for evaluation: 56 patients had completed the questionnaires for all three timepoints (TP1-3) (before, 6 and 12 months after surgery), 26 before and 6 months after surgery and 13 before and 12 months after surgery. The group of patients (*n =* 13), which only provided data for TP1 and 3 as well as *n =* 42 patients which only filled in TP1 were excluded from the evaluation. Seven guardians did not give their consent for their daughter to participate. Therefore, we have a total number of 82 included in the study.

54% of the patients suffered from type 1 and 46% from type 2 MRKHS. 

For the NeoCope questionnaire there were no corrections made for missing answers. In accordance with the FKB-20 manual not more than one value was imputated per scale. In case of a missing value it was replaced by the rounded item mean of the control group [23]. 

Equally, for the SESA rounded item means of the control group were imputated [27]. For PHQ-D, FSFI-d and SF-12 imputation was not used. 

Due to missing/incomplete data in the questionnaires the number of patients included in the following analyses differs.

### 3.2. NeoCope 

#### 3.2.1. Sociodemographic Data 

Data concerning age, height, weight, country of birth of patients and parents, actual living situation, highest educational grade and profession are given in Table 1. 

26% (*n =* 78) were from Southern Germany (zip codes 7 and 8).

#### 3.2.2. Diagnostic Procedures and Associated Malformations 

The MRKHS patients were mean 13.9 years (*n =* 75, *SD* 2.1, Mdn 15.3 years, Min 3.5, Max 20.0) old, when they themselves recognized there was something different with their body. MRKHS was diagnosed mean at 17.3 years (*n =* 81, *SD* 3.7, Mdn 16.6, Min 13.0, Max 34.0).

Primary amenorrhea was the main reason for gynecological examination in 93% (*n =* 82), followed by coital inability (22%) and the mother putting pressure on them (13%). In 4% the partner recognized a difference during petting, in 7% it was diagnosed during a routine check-up with a doctor, in 2% the patient found it during masturbation, in 2% it was diagnosed because of other malformations and in 7% there were other reasons for the examination.

Several examinations were performed in order to find the diagnosis: 76% (62/82) reported gynecological palpation, 79% (65/82) ultrasound, 57% (47/82) computed tomography (CT) or magnetic resonance imaging (MRI) and 54% (44/82) a hormone profile. In 2% (2/82) an examination under general anesthesia was performed and in 32% (26/82) a diagnostic laparoscopy. In 49% (36/74) the practicing gynecologist found the diagnosis, in 46% (34/74) a specialist at a hospital. 

More than half of our patients (61%, *n* = 71) reported not to be affected by associated malformations. Kidneys and lower urinary tract were the most commonly reported malformations (25%) followed by skeletal anomalies (11%). These answers are close to the preoperatively made classifications of 54% MRKHS type 1 and 46% type 2 (see above).

In addition to these malformations the patients reported a dystopic location of the heart, pelvic obliquity, club feet, blindness, syndactyly and a radius aplasia thrombocytopenia syndrome (TAR-syndrome). 

At TP1 (*n* = 78), TP2 (*n* = 80) and TP3 (*n* = 55) more than 40% of the patients did not report any cyclic discomfort. Preoperatively mainly Mittelschmerz was reported, followed by irritability, swollen abdomen and mastodynia. These symptoms remained after surgery but in a different order (Table 2).

#### 3.2.3. Laparoscopically Assisted Creation of a Neovagina

##### Age at Surgery

54% (*n* = 82) of the patients were minors when surgery was performed. The youngest patient was 15.3, the oldest 43.4 years old. Mean age at surgery was 20.0 years.

##### Reasons for Surgery

There were several reasons for surgery (*n* = 81). 65% had the wish to become a complete woman, 86% wanted to become sexually active, 17% wished to be able to have normal orgasms, 52% wanted to be seen by their partner like every other woman, in 1% the partner and in 3% the mother put pressure on the patient and in 15% there were other reasons for the surgery. Additional reasons were the wish to have normal partnership, the fear to lose the partner and the hope to feel mentally less disturbed after surgery.

The patients (*n* = 81) waited mean 29.7 months (SD 48.2, Mdn 12.0, Min 2.0, Max 348.0) between diagnosis and surgery. 

##### Self-Dilation and Preoperative Level of Information

Eleven patients performed self-dilation before deciding to undergo surgery. Six of them used the dilators for one hour daily (duration of self-dilation unknown), for the others data is missing.

87% (*n* = 82) of the patients have never tried to self-dilate. 

At TP2 76/82 patients (93%) said that they had been sufficiently informed about the procedure preoperatively. There was enough time to ask questions during the educational conversation with the doctor. The majority of patients had used the internet for more information (www.neovagina.de accessed on 16 March 2021) and some made contact with others affected personally or via mrkh-forum.com (not accessible anymore).

The patients were informed about the therapeutic options at our center via internet (TP2: 50%), by other affected women (TP2: 10%), by the practicing gynecologist (TP2: 45%) or a gynecologist at a hospital (TP2: 39%). Nevertheless, 26% (total number *n* = 77) at TP2 and 42% (total number *n* = 53) at TP3 complained about not having received sufficient information concerning MRKHS from their practicing gynecologist (Appendix A)

##### Surgical Outcome

Concerning the satisfaction with the surgical outcome the patients gave a mean score of 9.2 (SD 1.3, Mdn 10, Min 1, Max 10) at TP2 (*n* = 82) and of 9.5 (SD 1.2, Mdn 10, Min 3, Max 10) at TP3 (*n* = 55) on a scale between 1 (not at all satisfied) and 10 (very satisfied).

The medical care at our center was evaluated with 9.2 (SD 1.3, Mdn 10, Min 2, Max 10) at TP2 (*n* = 82) and 9.4 (SD 0.9, Mdn 10, Min 7, Max 10) at TP3 (*n* = 55) on a scale between 1 (very bad) and 10 (very good). 

Retrospectively, the patients rated the pain on the intermediate care unit with the traction device in place with a mean of 6.0 (SD 2.8, Mdn 6.8, Min 0, Max 10) at TP2 (*n* = 80) on a scale between 1 and 10. Nevertheless, 75 patients (92%, *n* = 82) would decide again for the surgical procedure and 81 patients (98.8%, *n* = 82) would recommend the method to others. 

##### Vaginal Stent

In order to maintain the surgical outcome, 77 patients (94%, *n* = 82) asked at TP2 had worn the vaginal stent during mean 4.2 months (SD 2.0, Mdn 3.0, Min 0.0, Max 9.0) and 37 patients (88%, *n* = 42) at TP3 had worn it during mean 6.7 months (SD 4.2, Mdn 6.0, Min 1.0, Max 18.0).

At TP2 35% (29/82) reported pain in the beginning, 56% (46/82) sometimes and 5% (4/82) always using the stent. In 43% (35/81) the stent induced bleeding in the beginning, in 51% (41/81) sometimes.

##### Behavior towards Men, Partnership and Sexuality

Of the patients 31% (TP2, *n* = 81) and 32% (TP3, *n* = 54) report an increased self-esteem. 

Preoperatively 39% of the patients (*n* = 74) were sexually active. The number increased at TP3 up to 84% (*n* = 55). 61% (*n* = 74) did not have any sexual contacts before surgery, compared to 16% (*n* = 55) after surgery (Table 3).

At TP2 the patients reported the first sexual intercourse mean after 9.4 weeks (SD 6.6, Mdn 6.5, Min 2.0, Max 32.0) and sexual intercourse on a regular basis from mean 10.5 weeks on (SD 6.7, Mdn 8.0, Min 4.0, Max 32.0).

Satisfaction with their sexuality was preoperatively rated with mean 4.8 (SD 2.8, Mdn 5.0, Min 1.0, Max 10.0) on a scale between 1 and 10, increased until TP2 to 7.8 (SD 2.0, Mdn 8.0, Min 2.0, Max 10.0) and remained stable at TP3.

##### Mother-Daughter Relation

48% (*n* = 77) of the patients reported an unchanged mother-daughter relation at TP2, for 42% of the patients this relationship had improved. At TP3 in 58% of the *n* = 55 patients the relationship was unchanged and in 36% it had improved. 

On a scale between 1 and 10, the patients evaluated their relationship with mean 8.3 (SD 2.4, Mdn 9.0, Min 1.0, Max 10.0) at TP2 (*n* = 65) and with mean 7.8 (SD 2.7, Mdn 9.0, Min 1.0, Max 10.0) at TP3 (*n* = 55).

The patients thought the mothers to be burdened by their own MRKHS diagnosis by mean 6.6 (SD 2.8, Mdn 8.0, Min 1.0, Max 10.0) at TP2 (*n* = 79) and 6.0 (SD 2.6, Mdn 6.0, Min 1.0, Max 10.0) at TP12 (*n* = 54). 

##### Relationship to Others

To speak about the MRKHS diagnosis with others is very difficult. Asked about whom they involved the patients named different persons of trust. At TP2 76% (62/82) talked to the mother about the diagnosis, 43% (35/82) to the father, 13% (11/82) to a brother, 29% (24/82) to a sister, 50% (41/82) to the partner, 49% (4/82) to friends, 22% (18/82) to others and 6% (5/82) to no one (Appendix A)

At TP2 45% (37/82) had contacted others affected, 24% (20/82) were contacted by others affected. Around half of the MRKHS patients does not have and does not want to have contact to others affected. Although 85% of the patients who have contact, find this helpful twelve months postoperatively (*n* = 33) and 93% six months postoperatively (*n* = 42). 67% of the patients who did not have contact with others affected, did not want to have this type of contact 6 months after surgery (*n* = 39) and still 59% twelve months postoperatively (*n* = 27). 

#### 3.2.4. Psychological Aspects

##### Feelings at Time of Diagnosis

Multiple selections were possible. The strongest feeling described by the patients (*n* = 82 at TP1 and 2) is the shock about infertility (61% at both TP). Feelings like desperation (45% at TP1 and 49% at TP2), emptiness, the question why them, unfairness, unstableness and sorrow have been described. 49% at TP1 and 55% at TP2 stated not to be able to accept the diagnosis and 28% and 31% not to be a normal woman due to the missing vagina. They feared to lose the partner or not to find one (26% at TP1 and 35% at TP2) due to the diagnosis. Other feelings were described in 13% at both TP.

##### Causes for Stress

Concerning the grade of stress due to the MRKHS diagnosis the patients reported a mean value of 6.5 (SD 2.3, Mdn 7.0, Min 1.0, Max 10.0) at TP1 (*n* = 80), 5.5 (SD 2.4, Mdn 5.5, Min 1.0, Max 10.0) at TP2 (*n* = 82) and 4.8 (SD 2.5, Mdn 4.0, Min 1.0, Max 10.0) at TP3 (*n* = 55) on a scale between 1 and 10. 

Infertility is one of the main issues at all three timepoints and is reported by 96% of the patients at TP3 (Table 4).

The infertility issue increases in relevance over time (Figure 1). 

Figure 1 shows the grade of stress due to infertility over time on a scale between 1 and 10. 

##### Psychosomatic/Psychotherapeutic Care

More than two thirds of the patients did not have any psychosomatic or psychotherapeutic support outside our department. Part of our patients treated at our centre were supported by the team of specialized psychologists from the Department of Psychosomatic Medicine. These patients evaluated the support with mean values of 6.6 (SD 2.8, Mdn 7.0, Min 1.0, Max 10.0) at TP2 (*n* = 25) and 6.3 (SD 2.8, Mdn 7.0, Min 1.0, Max 10.0) at TP3 (*n* = 11). 

### 3.3. PHQ-D: Psychological Disorders

#### 3.3.1. Continuous Scales—Characteristics, Reference Sample, Descriptive Evaluation

For the evaluation of the PHQ-D we first focused on the three continuous scales: somatic symptoms, depression and stress. 

The reference sample for continuous scales is according to Gräfe et al. mean 41.9 years old (SD 13.8) and includes 357 healthy participants and 117 patients with a mental disorder. 68% of the whole sample is female [28], while the MRKHS patients are obviously all female and have a mean age of 19.9 years (SD 5.3) at TP1, 20.6 years (SD 5.3) at TP2 and 21.4 years (SD 4.6) at TP3. To our knowledge there is no better fitting reference sample. 

The mean value for the somatic symptoms was between 4.8 and 5.4 for our MRKHS patients at all timepoints (Table 5). Compared to the reference sample there are no signs for more somatic symptoms in the MRKHS patients (Table 5). 

On the depression scale there was a value of 5.6 preoperatively which means mild/subliminal depression (values between 5 and 10). Postoperatively the values were within the normal range below 5. Concerning to Gräfe et al. the cut off for non-depressives is below 5.9, the sum score for patients with all depressive disorders is given as 11.7 (SD 5.0) [28].

For the stress scale with mean values between 4.2–4.4 in our cohort there are no comparative values given from the reference sample by Gräfe et al. 

Table 5 shows the descriptive evaluation of the continuous scales. Due to not normally distributed data we used the Wilcoxon signed-rank test. In the underlying literature by Gräfe et al. only arithmetic mean is given and not the median for somatic symptoms and depression. We suppose that the reference sample was big enough and normally distributed and that mean and median coincide. For statistical analysis the directional hypothesis test was used. Compared to the healthy reference sample there is no evidence for a somatization disorder in the MRKHS patients. On the other hand, the MRKHS patients differ significantly from the reference sample with mental disorders on both scales and at all timepoints, which means a lack of a depressive or somatic disorder (Table 5). 

##### Changes over the Time

Somatic symptoms

The scale for somatic symptoms includes the 15 most common somatic disorders, which indicate according to the DSM-IV (Diagnostic and Statistical Manual of Mental Disorders version IV) a somatization disorder. There was a strong correlation reported with the health-related quality of life [29].

Preoperatively, 16% showed a moderate to severe somatization, at TP2 still 10%, increasing until TP3 up to 20% (Table 6).

For the evaluation of changes over the time we used the Friedman test. As the Friedman-test (χ^2^ (2) = 1.624, *p* = 0.444) was not significant, a constant trend concerning somatic symptom severity cannot be excluded and a change not proven. The Conover’s Post-Hoc-test confirmed that there are no significant differences between the three timepoints.

Depression

The distribution of the cumulative values of the PHQ-8 for depression and their frequencies are shown in Table 6. The rate of mild and subliminal depressive disorders (values between 5 and 10) as well as the rate of major depression (values ≥ 10) nearly halve until TP2 and nearly remain constant until TP3. 

The scale for depression was analyzed over the course of time. Because the Friedman-test (χ^2^ (2) = 12.077, *p* = 0.002) became significant with a strong effect (Kendall’s W > 0.3) it can be emanated that there was a change over time. The Conover’s Post-Hoc test proved the differences between the timepoints. 

For the factor time on the scale for depression there was a significant difference pre- and six as well as pre- and twelve months postoperatively, which means a reduction of the mild depressive symptoms at the beginning over time. 

#### 3.3.2. Categorial Scales—Characteristics, Reference Sample, Descriptive Evaluation

The summary of the analysis of the categorial scales of the PHQ-D is given in Table 7. It shows the frequency of signs for mental disorders at the different timepoints and in comparison to the prevalence in the general population [30]. 

The reference sample for categorial scales according to Jacobi et al. has an age span of 18 to 34 years and includes only females, while the MRKHS patients have a much lower age span of 19 to 21 years of age. To our knowledge there is no better fitting reference sample. 

Signs for mental disorders in the majority of cases were less after surgery. Exceptions are only the signs for eating disorders (binge eating and Bulimia nervosa), which affected less patients before surgery and the somatoform syndrome (Table 7). 

We used the Chi-square test for equal distribution in order to compare the results of the categorial scales of the MRKHS patients with the reference sample by Jacobi et al. [30]. Significant differences can be drawn concerning somatoform (χ^2^ (1) = 8.712, *p* = 0.004) and major depressive syndromes (χ^2^ (1) = 4.755, *p* = 0.029) at TP2. In contrast to the assumption the MRKHS patient group shows a lower value than the reference sample. The alcohol syndrome shows preoperatively (χ^2^ (1) = 156.017, *p* < 0.001) a significant deviation. Eating disorders were not compared because of the impossible exact allocation to binge eating or bulimia nervosa (Table 7). 

### 3.4. FKB-20: Body Image

The self- and body-concept can be massively upset by an organic disease [23]. In order to be able to assess whether the surgery has changed the body image of the MRKHS patients, the course of the scale values of the rejecting body evaluation (RBI) and the vital body dynamics (VBD) was evaluated. 

According to Clement et al., the first reference sample of the FKB-20 is in average 23.9 years old (SD 3.3) and consists out of 141 medical students (40% female, 60% male). Only the data of 56 female medical students can be compared to our cohort. Additionally, a second reference sample with an average age of 32.6 years (SD 10.9), consisting out of 405 patients (63% female, 38% male) was selected. A comparison was possible with 253 female patients [23]. The patients from the FKB 20 manual show a broad spectrum of diagnoses and symptoms, e.g., eating disorders, transsexualism, personality disorders [22]. 

#### 3.4.1. Rejecting Body Evaluation (RBI)

##### Descriptive Evaluation

The upper limit for the scale of rejecting body evaluation (RBI) is 29. Values ≥ 29 indicate a body image disturbance or a negative body image. 27% of the MRKHS patients at TP1 (*n* = 82), 18% at TP2 (*n* = 80) and 18% at TP3 (*n* = 55) have a negative body image (Figure 2). Higher values indicate a more rejective body evaluation.

Data for all three timepoints are compared to both reference samples.

Due to missing normal distribution the Wilcoxon signed-rank test was used. Concerning the statistical analysis, a directed hypothesis test was chosen, because the median was in between both reference samples (Table 8). 

##### Changes over Time

Because of lacking normal distribution, the Friedman-test was performed. 

The Friedman test (χ^2^ (2) = 0.915, *p* = 0.633) is not statistically significant, which means that a change in the rejecting body evaluation cannot be shown and a persistent status cannot be excluded. The Conover’s Post Hoc test confirms no change between the different timepoints.

#### 3.4.2. Vital Body Dynamics (VBD)

##### Descriptive Evaluation

The VBD scale has a lower limit of 31. Higher scores indicate a better vital body image. Values below show a body image disturbance.

35% (*n* = 82) of the patients show preoperatively, 37% (*n* = 78) at TP2 and 31% (*n* = 55) at TP3 a value < 31 and therefore have signs of a body image disturbance (Figure 3).

The results of the VBD scale are compared to the RBI scale normally distributed at TP2 and 3 and therefore the Student-*t* test can be applied Table 8. To facilitate the analysis, it is also calculated for TP1. 

At every timepoint the null hypothesis had to be refused: MRKHS patients are significantly below the medical students and over the mean of the patient sample. As only mean values < 31 are seen as cut-off for a body image disturbance, the MRKHS patients are not affected (Table 8).

##### Changes over Time

Because there was normal distribution at TP2 and 3 a univariate ANOVA was used for the comparison over time. Concerning time there was no significant effect in VBD over the three timepoints: *F* (2, 102) = 0.283, *p* > 0.05, η^2^ = 0.001.

#### 3.4.3. Body Image Disturbance

In 18% of the MRKHS patients (*n* = 82) there was a value which indicated a body image disturbance preoperatively in both scales (RBI and VBD), in additional 27% in one of the scales. At TP2 still 14% (*n* = 78) and at TP3 13% (*n* = 55) of the MRKHS patients indicated body image disturbance in both scales.

### 3.5. FSFI-d: Sexual Function

With help of the FSFI-d it should be shown whether the patients were more limited in their sexual function preoperatively compared to postoperatively. For each single score the maximal value is 6 (total score 36). The higher the scores the better the sexual function. 

There is a major difference in terms of age between our MRKHS patients and the reference sample by Rosen et al. with an average age of 39.7 years (SD 13.2, Min 21, Max 68).

#### 3.5.1. FSFI-d Single Scores

Table 9 summarizes the descriptive analysis of the domains (desire, arousal, lubrication, orgasm, global satisfaction, and pain), which reached the MRKHS patients at the three timepoints. Each score is multiplicated with a loading factor taken from the literature in order to be better comparable with Rosen´s reference sample (desire (*0.6), arousal (*0.3), lubrication (*0.3), orgasm (*0.4), global satisfaction (*0.4), and pain (*0.4)) [24].

Descriptively, scores are lower preoperatively compared to six and twelve months postoperatively, indicating a stronger limitation in sexual function before surgery. In all domains sexual function improves postoperatively even reaching a higher score for global satisfaction than in Rosen´s reference sample at TP3 (Table 9). The Wilcoxon signed-rank test was again used because of not normally distributed data. Rosen et al. have only shown the arithmetic mean. However, as mentioned before we think the reference sample to be big enough for median and mean being mainly the same. For statistical analysis a directional hypothesis test is used after extensive literature search.

The MRKHS patients show stronger limitations within the domains desire, arousal and orgasm compared to the reference sample (Table 9). The domain lubrication shows a significant restriction (*p* > 0.05) at TP2 which disappears until TP3. Although the score for the domain pain increases at TP3 (*p* > 0.05), the domain for global satisfaction increases six months postoperatively (*p* < 0.05) (Table 9). 

#### 3.5.2. FSFI-d Total Score

Table 9 summarizes the FSFI-d total scores in the bottom part at all three timepoints. Normal distribution is missing at all timepoints (Shapiro-Wilk test, *p* < 0.05). A relevant increase in means and medians from pre- to postoperatively can be noticed. At all three timepoints the total FSFI-score is (although clearly increasing in the patients’ sample) still lower than in the reference sample. Figure 4 shows the distribution of the FSFI-d total scores at all three TP.

In Table 10 the FSFI-d total score was compared to the cut-off values by Wiegel et al. and Communal et al. [31,32]. 

Taking into account the cut-off value of 26.55 by Wiegel et al. we could show that the score of 19.0 strongly showed sexual dysfunction preoperatively and normalizes after surgery (Table 9).

Communal et al. (2003) categorized the total score into three severity codes: ≤23 bad/severe sexual dysfunction, 24–29 as good or satisfying/moderate sexual dysfunction, ≥30 as very good/normal sexual dysfunction [32]. 63% of the MRKHS patients (*n* = 41) show a severe sexual dysfunction preoperatively, decreasing down to 11% (*n* = 38) at TP3. 

#### 3.5.3. Changes over the Time for the FSFI-d Domain and Total Scores

In order to detect changes over the time and due to the missing normal distribution the Friedman-test was used. 

The Friedman-test became significant for all domains with a strong effect (Kendall’s W > 0.3): χ^2^ (2) > 6.608, *p* < 0.05 and therefore there are alterations over the time. 

The domains desire, arousal and satisfaction indicate a significant improvement in the Conover’s Post-Hoc test between TP1 and 2, also after Bonferroni correction (*p* < 0.05) (data not shown). 

For the domain pain there is a significant melioration over the course of time (data not shown). 

Concerning the FSFI-d total score there was a significant improvement (Friedman-test) over the time (*p* < 0.05) with a strong effect (Kendall’s W > 0.3). The difference between pre- and postoperatively was significant.

### 3.6. SF-12: Health-Related Quality of Life

The data of the MRKHS patients are compared to four reference sample groups, two healthy ones and two with acute or chronic diseases, each stratified by age (14–20 and 21–30 years of age). The reference sample groups include females and males.

#### 3.6.1. The Physical Component Summary Score (PCS)

The descriptive analysis and Wilcoxon signed-rank test of the physical component summary score (PCS) for all three timepoints is given in Table 11. Higher values indicate a better condition. Normal distribution is missing at all three timepoints (Shapiro-Wilk test, *p* < 0.05). Compared to the healthy reference sample of the 14–20-year-olds our data is unremarkable. 

The Wilcoxon signed-rank test shows significant better values for the physical component summary score (PCS) at all three timepoints in MRKHS patients compared to the reference sample *(p* < 0.05). This indicates that MRKHS patients do not differ from the reference sample.

#### 3.6.2. Mental Health Component Summary Score (MCS)

Data of the MCS are given in Table 11. Higher values indicate a better condition. Normal distribution is missing at all three timepoints (Shapiro-Wilk test, *p* < 0.05). Compared to the reference sample of similar age the values are lower at all timepoints. 

The Wilcoxon signed-rank test shows a significant deviation towards lower values of the MCS between the MRKHS patients and both healthy reference samples at all three timepoints and the acute and chronically ill reference sample for the preoperative values (*p* < 0.05). This indicates a significant impairment of the health-related mental quality of life. Postoperatively, there is no significant difference compared to the acute or chronically ill reference samples (Table 11).

Due to missing normal distribution the Friedman-test was used. 

There was no significant change over the time concerning PCS (χ^2^ (2) = 2.263, *p* = 0.322) and MCS (χ^2^ (2) = 3.677, *p* = 0.159).

### 3.7. SESA: Self-Acceptance Registration Scale 

The descriptive values at all three timepoints are shown in Table 12 and are compared to a healthy and a depressive reference sample [27]. The median of the healthy reference sample is 29 years of age consisting out of males and females. The median of the depressive reference sample is 45 years of age (28 females and 17 males). 

#### 3.7.1. SESA Total Score

The total score of the SESA was used in order to assess self-acceptance and self-esteem of the MRKHS patients (Table 12). Nearly for all of the depressive disorders a reduced self-acceptance is known. At TP2 and 3 normal distribution is missing, therefore we decided to perform the Wilcoxon signed-rank test.

Significant indicators for a worse self-esteem of the MRKHS patients compared to the healthy reference sample are found for TP1 (*p* < 0.05) but not for TP2 and 3 (Table 12).

The comparison with the depressive reference sample shows better values for the MRKHS patients for all three timepoints (Table 12). 

There were no significant changes concerning self-esteem over the time (Friedman-test). 

#### 3.7.2. Indicators for the SESA Total Score

A total score between 29 and 80 indicates a low self-acceptance, a score between 81 and 96 moderate self-acceptance and a score between 97 and 145 a normal self-acceptance. 

For more than 25% of the patients a moderate or low self-acceptance is obvious at all three timepoints (Table 13). 

The indicator for moderate and low self-acceptance were analyzed in more detail below. Patients which are not included in Table 14 show unremarkable values with a SESA total score (≥97).

Additionally, the MRKHS patients were individually evaluated at all three timepoints for indicators for low self-acceptance: eight patients show abnormalities with the total scores at all three timepoints (patients no. 3, 5, 9, 18, 36, 38, 43 and 54): for one patient (pat. No. 3) there were indicators for moderate self-acceptance at all three timepoints. The evaluations of the SESA questionnaires of patients No. 38 and No. 43 indicate low self-acceptance at all three timepoints. For patients No. 5 and No. 9 the total score decreases with time. The SESA total score of patients No. 18, No. 36 and No. 48 improves with time (Table 14). 

## 4. Discussion

### 4.1. Summary of the Aims and Instruments of the Study 

To our knowledge, this is the first study including a relevant number of MRKHS patients at three timepoints within a year and their long-term results concerning physical and mental health as well as sexual function after laparoscopically assisted creation of a neovagina in a standardized manner. The aim of the study was to confirm or disprove the following hypotheses among others:

MRKHS patients report a relevant mental and physical burden because of their malformation preoperatively.

The surgical therapy has a positive effect.

The patients confirm an improved sexual function and self-esteem after surgery.

Over the time there is a positive trend in mental coping with the anomaly.

Infertility remains an unsolved and relevant issue over the time.

At the three timepoints, which usually fall into the period of life between puberty and adulthood, when the development of a sexual identity as part of finding the own identity takes place, selected questionnaires were distributed. These consisted out of the following six questionnaires at TP 1, 2 and 3: Our own developed NeoCope questionnaire (see Appendix A) includes questions about demographic data, how and when diagnosis was made as well as the preoperative, peri- and postoperative situation. The development of the own body image (FKB-20), of the sexuality (FSFI-d), the subjective quality of life (SF-12) and the self-acceptance (SESA) in relation to MRKHS are evaluated in detail. Additionally, a screening questionnaire was used, in order to detect indicators for mental disorders (PHQ-D). 

#### 4.1.1. General

At the department for women’s health and the center for rare female genital malformations a high number of patients with this rare disease is treated annually, and therefore data of 56 patients was available before, six ad twelve months after surgery and additional 26 datasets before and six months after surgery. 

In most preceding studies not more than 20 patients could be included [33,34].

With data of 82 patients, this study provides valid and representative results for future studies in spite of the rareness of the disease. It can thereby help to optimize and improve the diagnostic and therapeutic procedures in MRKHS patients. All our patients were treated with the same procedure after a standardized diagnostic pathway and the same kind of information was distributed to all patients.

The data was collected prospectively within the clinical routine, a fact that implies also problems for the recruiting of patients like changing physicians, a huge catchment area (only 26% of the patients were from Southern Germany) and patients that only presented for surgery and never for follow-up visits. In general, like with other malformations, MRKHS patients do not want to be reminded all the time of being different, but want to live a normal life as soon as possible after therapy. 

Concerning the return of the questionnaires and the willingness to participate in the study there are different further aspects: as the majority of the patients are minors at the time of surgery, at least one parent has to give its consent. On the other hand, the package consists of six questionnaires for each timepoint, completing is time consuming and includes very personal topics. Therefore, it is understandable that 4% (*n* = 160) of the parents did not consent and an additional 31 % (*n* = 137) of the MRKHS patients had to be excluded due to incomplete questionnaires.

The sociodemographic data gives a cross section through the society related to a female group with an average age between 19.9 and 21.4 years. Most of the questionnaires have been validated with an older reference sample.

#### 4.1.2. Mental Stress of the MRKHS Patients 

The study confirms the preoperative inability to cohabitate and at all three timepoints the infertility as significant causes for mental stress. In the following sections we will discuss our data in relation to the four critical timepoints for psychological stress according to Bean et al. [7].

##### Critical Timepoint 1: Finding and Informing about the Diagnosis

Primary amenorrhea was the reason leading to the diagnosis of MRKHS in 93% of our patients. Patients were in average 13.9 years of age, when they recognized something was different with their body, but diagnosis was made in average only with the age of 17.3 years. Thereby, the MRKHS patients are confronted with an insecure period of 3.4 years during puberty, when they are usually supposed to develop their own female identity starting at the time of menarche. Additionally, the risk of primary misdiagnosis is high and was described with 41% [35], leading to the development of a diagnostic algorithm in 2008 [36]. The application of the suggested diagnostic algorithm is confirmed by this study. Although, we could detect that the preoperative amount of information was sparse with 55% of the gynecologists being able to only give incomplete or no detailed information. 

Extragenital associated malformations mainly of the kidneys and lower urinary tract and the skeletal system were equally often as described in the literature [5,37,38]. 

Besides the diagnostic and medical aspects, being informed about the diagnosis is synonymous with the shock of a traumatizing event [39]. 61% of the MRKHS patients stated to feel inferior and shocked by the infertility, followed by the fact to disavow the diagnosis (49%) and huge desperation (45%). Inner blankness, insecurity, helplessness and grief are also described. Independent from the therapeutic method, other authors have recommended and interdisciplinary and empathic approach for MRKHS patients [8,34,40] and to offer a psychological treatment in all cases [18].

It can be problematic when the diagnosis is communicated by a not psychologically skilled person and without taking enough time. This is not the case for our patients as we have many years of experience with these patients and psychological support has been established more than 15 years ago routinely or on demand. The individual personal situation and support by family and friends should be taken into account and an early follow-up appointment offered. 

##### Critical Timepoint 2: Surgical Creation of a Neovagina

Since 1988 the laparoscopically assisted creation of a neovagina (modified Vecchietti technique) is successfully performed at the Department for Women’s Health in Tübingen. This state-of-the-art procedure enables the mainly adolescent MRKHS patients to have a normal sexually active life. 

The main reason for surgery was the wish to be able to cohabitate (86%), followed by the wish to feel like a complete and normal woman (65%) and to be seen by the sexual partner like any other woman (52%). 54% of the MRKHS patients were minors at the time of surgery (*n* = 82). The youngest patient who underwent surgery was 15.3, the oldest 43.4 years of age and the average age at surgery was 20 years. In average the patients waited 29.7 months between receiving the diagnosis and surgical therapy. 87% of the patients did not use self-dilation before surgery. Overall, the MRKHS patients are very satisfied with the surgery, the hospital care and the functional results, similar to the literature [14,41,42,43]. In order to receive the postoperative result 94% wore the phantom for 4.2 months postoperatively. Retrospectively, the postoperative pain on the intermediate care unit with the traction device in place was rated with 6 but tolerable in view of the success of surgery. Nevertheless, the possible postoperative pain situation should be explained and prepared extensively in advance. The use of a pain diary may be an option in order to optimize the monitoring of the pain after surgery. Already preoperatively, the management of former critical situations and individual coping strategies should be detected. 

The postoperative measures were accepted and guaranteed the surgical success, which made sexual intercourse possible after four weeks. In average after 10.4 weeks our patients initiated regular sexual intercourse and 99 % of the MRKHS patients would recommend the method to others. 

Mental disorders pre- and postoperatively (PHQ-D)

The stress caused by MRKHS did not lead to a higher risk for mental disorders in our patients compared to a healthy reference sample and our patients were better off compared to a mentally ill reference sample. There was no aggravation or improvement concerning somatic symptoms over the time. Nevertheless, 16% of the patients showed preoperatively and 20% twelve months postoperatively a moderate to severe somatization.

The symptoms of a somatization disorder are not intentionally provoked and are often associated with the coping of a disease. Bean et al. reported that the language used can influence the experience of a woman with MRKHS positively or negatively [7]. Our specialized team of gynecologists is empathic and alert concerning psychosomatic symptoms which might develop in MRKHS patients over the time. The patients are offered psychosocial support at an early moment and not only focused on the physical anomaly. The early education that the diagnosis itself and especially the associated infertility will need a coping process, prevents a somatization disorder from developing. 

Depressive symptoms

A depressive disorder is one of the commonest mental disorders implying relevant consequences. It is a collective term for different characteristics and diseases. In the study for adult health in Germany (DEGS1) it was found that the prevalence of depressive symptoms is age-dependent and that women are with 10% more often affected than men with 6%. Additionally, the prevalence is highest in the age group of 18–19 years of age (12%) and decreases thereafter [44].

Our patient cohort belongs to a high-risk group for developing depressive symptoms due to the MRKHS diagnosis, the age between 18–29, being female and infertile. Hypothetically, we assumed persisting depressive values not only preoperatively but also postoperatively, because Langer et al. described 36% (*n* = 11) MRKHS patients being depressive postoperatively [45]. Möbus et al. identified higher depressive scores even after a successful creation of a neovagina in 44 MRKHS patients [46]. The results were explained by the persisting infertility. Epstein and Rosenberg had proven a significantly higher depressive stress due to primary infertility [47] and recently Chen et al. documented depressive symptoms even in 75% of the examined MRKHS patients as well as in one third of the patients a higher risk for depressive disorders [48].

Different from the literature and because of the small sample size, with the PHQ-D we detected a significant mild/subliminal depression preoperatively, with postoperatively no higher prevalence compared to the general population [44]. Using the SESA questionnaire, nearly 70% of the MRKHS patients did not need a differentiated examination in order to exclude a depressive disorder at any of the timepoints and compared to a depressive reference sample our patients showed significant better scores. For about 25% of our patients a differentiated examination was either necessary or an indicator for a depressive disorder was already present. The reduction of the mild depression from the beginning until TP2 and 3 indicates an initiated coping concerning the psychological abnormalities (inactivity, desperation, depressiveness). Even the prevalence of a major depression decreases postoperatively from 33% to 15% indicating a positive effect of the laparoscopically assisted creation of a neovagina and the thereby possible fulfilled partnership. Additionally, the confrontation with infertility at an early age makes it possible to deal with this issue earlier than usual, which can be an advantage and raise the chances of e.g., a successful adoption. Nevertheless, screening instruments like the PHQ-D should be used regularly in order to be able to support the patients as soon as possible when necessary especially when childbearing becomes an issue in the late twenties.

Psychological anomalies: eating disorders and alcohol syndrome

Descriptively, in the majority of cases our study showed less indicators for psychological disorders after surgery. Compared to the general population (females between 18 and 34 years of age) there was no higher prevalence of psychological disorders except for eating disorders (mainly bulimia nervosa and binge eating) and the alcohol syndrome. Heller-Boersma et al. also found conspicuous values of an eating disorder [18]. Langer et al. reported the overlap of the MRKHS diagnosis and an abnormal body image [45]. Eating disorders are known to be associated with disturbed body perceptions and therefore the malformation can increase the risk for the development of an eating disorder. As our study participants are mean between 19.9 and 21.4 years of age, they are within a period of life when they find and develop an own body image without MRKHS necessarily influencing this process. However, as the mentioned types of eating disorders are known to serve to deal with unpleasant emotions like stress, anxiety and desperation, which belong to the coping with MRKHS, patients should be offered psychotherapeutic help at any time. 

The alcohol syndrome can be seen as a normal sign of development of an own identity during puberty and adolescence and is not necessarily related to the MRKHS diagnosis. 

Body image disturbance pre-/postoperatively (FKB-20)

According to developmental psychology, adolescence with its physical changes and maturation crises has to be finished before a stable body image can develop. The mean values for the rejecting body evaluation (RBI) were for all three timepoints and the majority of the patients within the normal range and stable. 27% of the patients had a negative body image preoperatively, decreasing to 18% postoperatively. The vital body dynamics (VBD) shows similar results: still 31% of the MRKHS patients show a body image disturbance at TP3. An overlap on both scales (RBI and VBD) indicates a body image disturbance in 18% before and 13 to 14% after surgery. Compared to a patient sample with a wide range of symptoms and diagnoses MRKHS patients were significantly better.

Klingele et al. described 2003 in 55% of their patients a better self-image after surgery. Postoperatively, 62% were satisfied with their total body appearance [49]. Riessen et al. could show a significant improvement in body image six months after our procedure [50]. A stable body image was not possible yet to develop in our patients and adolescence itself certainly has a relevant impact on our findings. The laparoscopically assisted creation of a neovagina for sure has a positive effect, but a screening for body image disturbances is to be recommended and therapeutic options discussed. 

Health related quality of life pre-/postoperatively (SF-12) 

The diagnosis MRKHS poses a big challenge for the affected adolescents concerning their health-related quality of life, as even after creation of a neovagina they will not be able to carry a pregnancy. 

For physical health (PCS) the patients show better values than the reference sample of chronically ill and even of healthy patients. Data for mental health indicate a significant impairment caused by MRKHS in all four areas: vitality, social functionality, emotional role function, mental wellbeing. This finding is confirmed by the comparison with the reference sample of chronically ill patients. Although the impairment decreases after surgery it remains within a low level and significant compared to the chronically ill 14–20 year olds pre-, as well as the 21–30-year-olds at TP1 and 2. The health related quality of life of MRKHS patients so far was only evaluated in a small number of studies: Kaloo et al. did not detect any difference, Klingele et al. could detect an improvement in quality of life of the MRKHS patients compared to the general population in 79% and Keckstein et al. 2008 in general, but with small sample sizes [49,51,52]. Only Liao et al. differentiated between physical and mental health, reporting better values in physical and worse values in mental health [53]. 

MRKHS patients in our study seemed to be physically unstressed but mentally impaired, and the higher values on the PCS can be explained by a familiarization with the MRKHS diagnosis and the surgical therapy and the normal “operation” in daily life. On the contrary, chronically ill patients usually have a relevant physical impairment. 

Interesting is the fact, that MRKHS patients have a stronger mental impairment than chronically ill patients. This might be due to the fact that the SF-12 does not specifically distinguish between infertility and other reasons and therefore MRKHS patients seem to be more mentally impaired. The burden of infertility remains a focus throughout different phases of life, different from other chronic diseases which will be less and more relevant depending on the actual personal situation. The use of instruments other than the SF-12, which more specifically focus on quality of life can be useful in order to develop targeted coping strategies for MRKHS patients. 

Altered self-acceptance after surgical creation of a neovagina (SESA)

With the SESA questionnaire we focused on the course of time and its effect on the self-esteem. We hypothesized that MRKHS patients have a lower self-acceptance before compared to after surgery. The hypothesis could be confirmed as the patients preoperatively had a lower self-esteem compared to the healthy reference sample. After surgery there was a normal self-acceptance and patients did not significantly differ from the reference sample. This is in contrast to the common hypothesis that MRKHS patients have a reduced self-esteem in general. Literature concerning self-acceptance in MRKHS is diverse. Heller-Boersma et al. supposed doubts concerning the own identity, a lower self-acceptance and self-esteem as a result of the information about the diagnosis [8]. Morgan and Quint reported values within the average range [33]. Möbus et al. and Klingele et al. as well as Keckstein et al. showed a significantly improved self-esteem after surgery [46,49,52]. All these evaluations were performed retrospectively and did not include pre- and postoperative data. 

Besides the postoperatively better self-esteem, our patients could easier engage into a partnership after surgery and approach men with a better self-esteem. Patients see themselves more competent in coping with MRKHS in different situations. 

##### Critical Timepoint 3: Relationships and Sexuality

Impairment of the mother and mother-daughter-relationship 

The mother is the number one person of trust for 76% of our MRKHS patients. The unconditional love between mother and daughter is put to the test during puberty and the finding of the own identity. The latter makes discrimination and own experiences necessary, and the peer-group becomes more important. The mother often has the feeling of losing something. The MRKHS diagnosis for the majority of the patients falls into this vulnerable phase of life and the mother changes into a passive role. The stress intensity of the mother on a scale between 1 and 10 was reported by the patients with an average value of 6.6 at TP2 (*n* = 79) and of 6.0 at TP3 (*n* = 54). The mother-daughter-relationship is mainly rated as positive and unchanged over the course of time. There was no relevant pressure by the mother to undergo diagnostic procedures (only in 13.4%) or surgical creation of a neovagina (only in 2.5%).

It is known that the mother-daughter-relationship has a critical influence on the development of female sexuality and the body image. The positive regard through the mother can have a positive effect on the postoperative results and improve the coping [7,54]. In a study by Leithner et al. MRKHS patients (*n* = 10) describe their mothers as especially caring [34]. Because of the reported stress intensity of the mothers, support should be offered, in order to be able to process their emotions, to recognize grief and sense of guilt and finally be able to support their daughters adequately.

Social support

Yearly patient days at the Department for Women’s Health in Tübingen could prove that MRKHS patients benefit from the personal exchange with others affected. In different seminars and workshops not only patients, but also relatives, partners and friends can talk to each other. 

Ernst et al. reported that the contact and the communication about the MRKHS diagnosis among peers during adolescence under adequate guidance can be a source of support and strengthen relationships and self-esteem [55]. In addition to the patient day 45% of our patients had contact to others affected or patient groups and the majority found these contacts helpful [56]. For the motivation to tell others of the peer group about the diagnosis a strong confidence in the relationship is needed. The commonest reason not to tell others is the fear to meet with a refusal or be seen as a freak [45,57]. 5% of our patients did not take anyone into their confidence concerning the MRKHS diagnosis. It has been described before that some MRKHS patients wish to keep the diagnosis as their secret [40,57]. Nevertheless, 50% of the MRKHS patients share their worries with their family and friends twelve months after surgery. The removal of taboos and an open handling with the diagnosis are important factors. Patients must receive careful support concerning the topics how to describe and explain the malformation and to whom. Social support can thereby help to cope with the MRKHS diagnosis. 

Partnership and sexuality

The issues partnership and sexuality are in association with the laparoscopically assisted creation of a neovagina two critical aspects: after surgery the patients were more often within a stable relationship than before surgery. 31% were more open and self-confident towards men after surgery than before. In our study sexual activity before and after surgery could be compared. Ismail-Pratt et al. found an evaluation of the preoperative sexuality not useful, because patients were not sexually active [58]. We could not confirm this opinion, because 39% of our patients stated to have been sexually active before surgery with petting being the commonest form of sexual contact in 59%. The frequency of sexual contacts and the satisfaction with these increased constantly with the time from surgery. Moreover, the number of sexual partners increased. In total, the results can be seen as success as the patients develop their sexuality and dare to approach potential partners without too much fear. 

FSFI-d

By using the FSFI-d we were able to confirm previous studies concerning sexual function [20,34,42]: preoperatively it was, as expected, limited and normalized postoperatively after creation of a neovagina. Six and twelve months after surgery the scores lay within the normal range for the majority of patients. Concerning satisfaction patients even reached a significant higher score compared to a normal reference sample. A normal sexuality is confirmed by 40% of the MRKHS patients twelve months after surgery. All domains improve with the time from pre- to postoperatively. 

In comparison with the FSFI-d total score with the cut off value by Wiegel et al. 34% of the patients indicate sexual dysfunction twelve months after surgery. It is questionable if this means a dysfunctional neovagina or exaggerated expectations especially as both samples differ in age.

##### Critical Timepoint 4: Starting a Family (Infertility)

With the information about the MRKHS diagnosis patients were informed about infertility. The awareness comes with the time: preoperatively 78% (*n* = 82) concerned themselves with this issue, twelve months postoperatively already 96 % (*n* = 55). Graduation from school, finding a job, partnerships and the developing sexuality are in the front. After surgery the focus shifts with a stress intensity due to infertility of 7.5 on a scale between 1 and 10 at TP3. MRKHS patients are forced to deal with infertility much earlier compared to other women. In Germany the mean age at the birth of the first child is 29.8 years [59]. Normally, women in their end twenties think about starting a family and not in the early twenties. MRKHS patients are lacking life experience in order to deal with the existential issue of infertility in an outcome-oriented way. Though, the study by Fliegner et al. has shown an only moderate wish to have children, justified with the young age of the MRKHS patients (median 22.0 years of age). Additionally, there was an ambivalence reported concerning motherhood, which is interpreted as insecurity in the phase of self-discovery and the development of aims in life [60]. 

Jealousy of fertile women were described elsewhere when friends started a family [40]. 

Nevertheless, Stöbel-Richter et al. could show that in spite of infertility the majority of couples have an intact partnership and a high satisfaction with life [61]. 

Our results point to an emotional stress due to the infertility. Changing importance of this issue can be expected during different phases of life. MRKHS patients must be offered the best possible psychosocial and psychological support, not only to be able to find alternatives and individual ways of becoming parents. 

### 4.2. Strengths and Limitations of the Study

One strength is the voluntary participation independent from psychological support or surgical procedures. Within a year, data was evaluated at three different timepoints. The diversity of the used questionnaires with focus on different areas is exceptional. The diagnostic process and only MRKHS patients are included. Thereby, future optimization concerning a standardized diagnostic procedure and therapeutic concept can be a result. While all our patients were treated with the same procedure after a standardized diagnostic pathway and the same kind of information was distributed to all patients.

The laparoscopically assisted creation of a neovagina was elucidated in detail and was recommended by the majority of patients in spite of the postoperatively reported pain.

A unique feature of the study is the number of included patients in consideration of the rareness of the disease. 

Limitations are the self-reported questionnaires without any external objectivity. A digital completion and evaluation would be of advantage and might have resulted in more completed questionnaires. 

Due to the young age of our patients the burden of infertility could only partly be evaluated. Future follow-up studies are needed. 

There is little age-matched reference data for our patient cohort. It has to be taken into account that the reference samples are often much older and mixed-gender.

### 4.3. Perspective and Implications for the Support and Therapy of Women with MRKHS

In this study we could show the need for an early and correct diagnosis. The introduction of a routinely performed preventive gynecological check-up in girls between 12 and 14 years of age in order to rule out—so far asymptomatic—genital malformations may be one possibility. This appointment could be combined with an education about HPV and the recommendation of a prophylactic vaccination. 

Several questions showed up during our study: The sexual orientation in our cohort, the wording and setting during information about the diagnosis, e.g., was not part of the questionnaires. Of interest is also the aspect if MRKHS patients change their choice of occupation in context of the experienced infertility e.g., in order not to be confronted with children all the time. 

Useful could also be the characterization of the different personalities with special focus on their coping styles, while the subjective evaluation of stress factors can probably never be performed with standardized instruments, instead a new method would have to be developed. Some authors have already previously reported that a good adaptation is possible for patients with MRKHS and that after creation of a neovagina a reasonable handling with the malformation can be achieved [62]. Nevertheless, it would be of interest how open the issues gender identity and sexuality are discussed within a family including the father, its stress and willingness to support. The mother-daughter relation and its development over the time with the patients first steps into independence. Migrational and cultural backgrounds should also be taken into account. 

In future studies, the influence of uterus transplantation as a new option for MRKHS patients in several countries should be included and evaluated.

The offered support might not be suitable yet and patients cannot decide if it is useful or not.

After performing this study, we still do not think that a routinely psychosomatic/psychologic counselling for all patients is reasonable, but a routinely screening should be established. 

### 4.4. Conclusions

This is the first study with a decent number of MRKHS patients, which included besides sexual function several questionnaires covering health and quality of life before and after surgical therapy. 

In conclusion, we detected an impairment of the health-related mental quality of life but no automatically higher risk for psychological disorders caused by the diagnosis. 

MRKHS patients show a normal self-acceptance, lacking a body image disturbance in most cases. As assumed, the sexual function is limited before surgery and normalizes after surgery. 

Useful factors for coping with the disease are psychosocial adaptation as well as a supportive social environment. The contact to other affected young women has a positive effect on the development of coping strategies. A routine psychologic or psychosomatic counselling is not indicated but has to be balanced and provided individually as an interdisciplinary approach depending on patients’ individual needs.

## Figures and Tables

**Figure 1 jcm-10-01269-f001:**
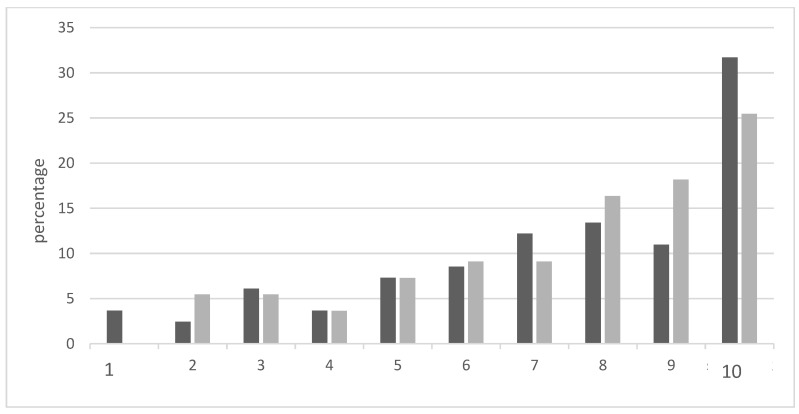
Grade of stress due to infertility. Dark grey bars represent TP2 (*n* = 82), the light grey bars TP3 (*n* = 55).

**Figure 2 jcm-10-01269-f002:**
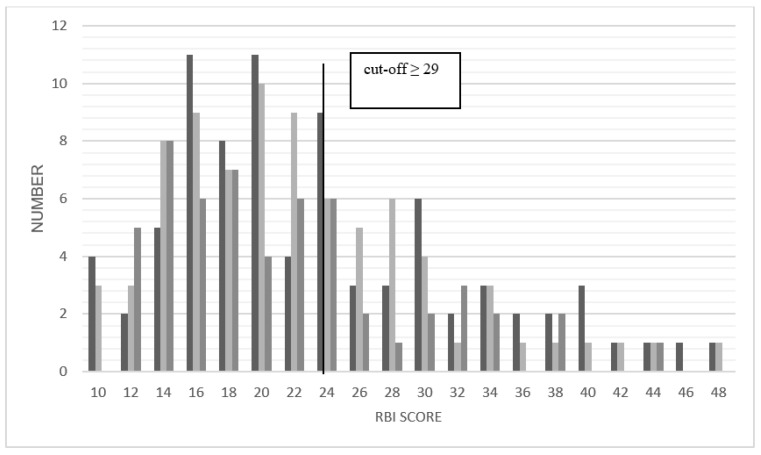
Overview over the distribution of the RBI scores (FKB-20). notes: black bars represent the RBI score at TP1 (*n =* 82), light grey bars at TP2 (*n =* 80) and dark grey bars at TP3 (*n =* 55).

**Figure 3 jcm-10-01269-f003:**
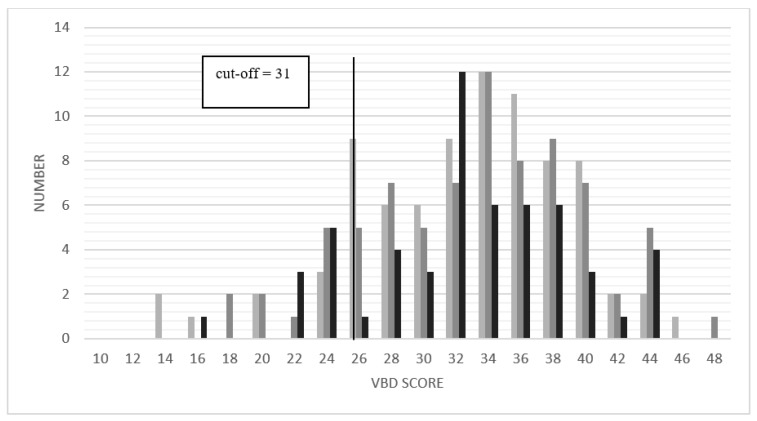
Overview over the VBD scores (FKB-20). notes: black bars represent the VBD score at TP1 (*n =* 82), light grey bars at TP2 (*n =* 78) and dark grey bars at TP3 (*n =* 55).

**Figure 4 jcm-10-01269-f004:**
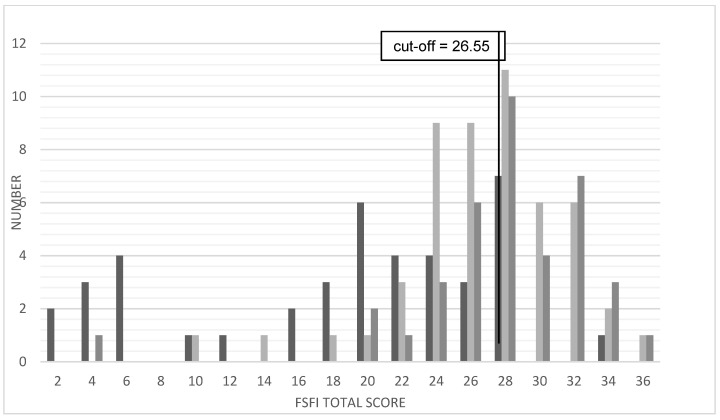
Distribution of the FSFI-d total scores. notes: black bars represent the FSFI-d total score at TP1 (*n =* 41), the light grey bars the FSFI-d total score at TP2 (*n =* 51) and the dark grey bars at TP3 (*n =* 38).

**Table 1 jcm-10-01269-t001:** Sociodemographic data of MRKHS patients (data collected TP1-3).

Demographic Data	TP1		TP2		TP3	
	*n*	M (SD)	*n*	M (SD)	*n*	M (SD)
Age (years)	82	19.9 (5.3) ^a^	82	20.6 (5.3) ^a^	56	21.4 (4.6) ^b^
Average height (cm)	82	166.8 (6.7)	82	167.2 (6.6)	56	167.1 (7.1)
Average weight (kg)	82	60.7 (11.9)	82	61.9 (12.1)	56	62.1 (12.3)
BMI (kg/m^2^)	82	21.3 (4.3)	82	22.2 (4.5)	56	22.6 (4.1)
	N	*n* (%)	N	*n* (%)	N	*n* (%)
Country of birth, patients	81					
Germany		73 (90%)				
other		8 (10%)				
Country of birth, parents	81					
Germany		65 (80%)				
One parent from foreign country		5 (6%)				
Both parents from foreign country		11 (14%)				
Number of siblings	79					
sisters						
0		37 (47%)				
1		35 (44%)				
2		7 (9%)				
brothers						
0		38 (48%)				
1		25 (32%)				
2		10 (13%)				
3		5 (6%)				
4		1 (1%)				
Living situation: I live	82		82		55	
with my parents		66 (81%)		65 (79%)		36 (66%)
alone in a flat		5 (6%)		9 (11%)		5 (9%)
with others in a shared flat		2 (2%)		3 (4%)		5 (9%)
with my girl-/boyfriend in a flat		9 (11%)		5 (6%)		9 (16%)
Highest school degree	73		81		55	
none		10 (14%)		5 (6%)		3 (5%)
secondary modern school		12 (16%)		12 (15%)		4 (7%)
junior high school		30 (41%)		30 (37%)		16 (29%)
final secondary school exam		17 (23%)		27 (33%)		29 (53%)
university degree		4 (6%)		7 (9%)		3 (6%)
Actual profession	78		82			55
unemployed		1 (1%)		2 (2%)		0
scholar		41 (53%)		36 (44%)		19 (35%)
trainee		13 (17%)		18 (22%)		15 (27%)
student		4 (5%)		6 (7%)		10 (18%)
employee		14 (18%)		15 (18%)		10 (18%)
officer		0		0		0
self-employed		0		0		0
others		5 (6%)		5 (6%)		1 (2%)

Notes: ^a^ age span from 17 and 40 years. ^b^ age span from 18 to 44 years.

**Table 2 jcm-10-01269-t002:** Cyclic complaints.

Complaints	TP1	TP2	TP3
Do you suffer from cyclic discomfort (regular monthly complaints for 1–3 days)? (Multiple selections possible)	*n* = 78	*n* = 80	*n* = 55
mastodynia	16 (21%)	16 (20%)	16 (29%)
Mittelschmerz/lower abdominal pain	30 (39%)	24 (30%)	13 (24%)
irritability	16 (21%)	24 (30%)	14 (26%)
discharge	4 (5%)	2 (3%)	8 (15%)
ravenousness	15 (19%)	18 (23%)	13 (24%)
swollen abdomen	16 (21%)	15 (19%)	10 (18%)
other cyclic complaints	8 (9%)	2 (3%)	3 (6%)
no complaints	34 (44%)	35 (44%)	23 (42%)

**Table 3 jcm-10-01269-t003:** Sexual contacts, sexual partners, sexual intercourse (different patient numbers show different rates of completed questions).

Sexuality	TP1	TP2	TP3
Are you actually sexually active?	*n* = 74	*n* = 82	*n* = 55
Have you ever had sexual intercourse?			
No	45 (61%)	28 (34%)	9 (16%)
yes	29 (39%)	54 (66%)	46 (84%)
Which type of sexual contact do you have?	n = 27		
petting	16 (59%)		
sexual intercourse	4 (15%)		
both	7 (26%)		
Did you have more than one sexual partner after surgery?		*n* = 52	*n* = 46
No		48 (92%)	27 (59%)
yes		4 (8%)	19 (41%)
Number of sexual partners since surgery		*n* = 5	*n* = 18
2		5 (100%)	9 (50%)
3		0	5 (28%)
4		0	3 (17%)
12		0	1 (6%)
How do you orgasm? (multiple selections possible)		*n* = 82	*n* = 56
(day)dreams		5 (6%)	3 (5%)
masturbation		23 (28%)	28 (50%)
petting		32 (39%)	30 (54%)
sexual intercourse without petting		17 (21%)	22 (39%)
sexual intercourse with petting		42 (51%)	35 (63%)
never		13 (16%)	5 (9%)

**Table 4 jcm-10-01269-t004:** Causes for stress associated with the MRKHS diagnosis.

Causes for Stress	TP1	TP2	TP3
Which topics stress you the most associated with MRKHS? (multiple selections possible)	*n* = 82	*n* = 82	*n* = 82
unwanted childlessness	64 (78%)	71 (87%)	53 (96%)
therapy/surgery	49 (60%)	16 (20%)	11 (20%)
wish for a fulfilled partnership	33 (40%)	35 (43%)	17 (31%)
wish for a satisfying sexual life	33 (40%)	40 (49%)	14 (26%)
my self-esteem	28 (34%)	24 (29%)	20 (36%)
mother-daughter relation	4 (5%)	6 (7%)	9 (16%)
to be physically different from my friends	31 (38%)	30 (37%)	24 (44%)
to be different from other women	23 (28%)	22 (27%)	17 (31%)
other topics	2 (2%)	2 (2%)	7 (13%)

**Table 5 jcm-10-01269-t005:** Descriptive evaluation and Wilcoxon signed-rank test of the continuous scales (PHQ-D) compared to the values from two reference samples.

Continuous Scales	n	M (SD)	Mdn	Min	Max	*p*-Value (TP1, TP2, TP3)
somatic symptoms						
TP1	81	5.4 (4.2)	5	0	16	
TP2	80	4.8 (4.0)	4	0	19	
TP3	55	5.4 (5.2)	4	0	28	
^1^ healthy	357	6.4 (3.9)				0.014 *, <0.001 *, 0.032 *
^2^ mental disorders	171	9.8 (5.4)				<0.001 *, <0.001 *, <0.001 *
depression						
TP1	81	5.6 (4.2)	5	0	17	
TP2	79	3.8 (3.8)	3	0	20	
TP3	55	4.0 (4.6)	3	0	24	
^1^ healthy	357	5.9 (4.2)				0.294, <0.001 *, <0.001 *
^2^ mental disorders	171	11.7 (5.0)				<0.001 *, <0.001 *, <0.001 *
stress						
TP1	81	4.4 (3.3)	4	0	14	
TP2	79	3.6 (3.1)	3	0	13	
TP3	55	4.2 (3.3)	4	0	15	

Notes: Wilcoxon signed-rank test. * *p*-value < 0.05. ^1,2^ reference sample: Gräfe et al. [28].

**Table 6 jcm-10-01269-t006:** Frequency and rates of the scores for different manifestations of the somatic symptoms value (PHQ-15) and frequency and rate of values for the different manifestations of depression (PHQ-8 and PHQ-9).

Scores and Values	TP1	TP2	TP3
Score of somatic symptoms	*n* = 81	*n* = 80	*n* = 55
0 to 4 (minimal)	40 (49%)	43 (54%)	30 (55%)
5 to 9 (mild)	28 (35%)	29 (36%)	14 (26%)
10 to 14 (moderate)	9 (11%)	5 (6%)	9 (16%)
15 to 30 (severe)	4 (5%)	3 (4%)	2 (4%)
Value for depression	*n* = 81	*n* = 79	*n* = 55
<5 (no depressive disorder)	35 (43%)	54 (68%)	36 (66%)
5 to 9 (mild or subliminal depressive disorder)	33 (41%)	19 (24%)	15 (27%)
≥10 (major depression)	13 (16%)	6 (8%)	4 (7%)
10 to 14 (moderate “major depression”)	11 (14%)	4 (5%)	2 (4%)
15 to 19 (pronounced “major depression”)	2 (3%)	1 (1%)	1 (2%)
20 to 24 (severe “major depression”)	0	1 (1%)	1 (2%)

**Table 7 jcm-10-01269-t007:** Frequency and rates of the categorial scales and Chi-square test for equal distribution for the comparison of the categorial scales with the reference sample by Jacobi et al. (PHQ-D).

Categorial Scales	TP1	TP2	TP3	Reference Sample	*p*-Value
				[30]	(TP1, TP2, TP3)
	***n* = 82**	***n* = 82**	*n* = 56		
somatoform syndrome	6 (7%)	3 (4%)	5 (9%)	15%	0.053, 0.004 *, 0.209
major depressive syndrome	4 (5%)	2 (2%)	3 (5%)	10%	0.153, 0.029 *, 0.290
other depressive syndromes	13 (16%)	3 (4%)	1 (2%)		
panic syndrome	2 (2%)	0	1 (2%)	3%	0.631, unk, 0.505
other anxiety syndromes	2 (2%)	2 (2%)	1 (2%)		
suspected Bulimia nervosa	1 (1%)	3 (4%)	1 (2%)	1% °	
suspected Binge eating disorder	4 (5%)	10 (12%)	4 (7%)		
alcohol syndrome	17 (21%)	5 (6%)	2 (4%)	2%	<0.001 *, 0.005, 0.359

Notes: * *p*-value < 0.05. ° suspected Bulimia nervosa and Binge eating disorder together.

**Table 8 jcm-10-01269-t008:** Descriptive evaluation and Wilcoxon signed-rank test of the RBI scale (FKB-20) compared to two reference sample groups and descriptive analysis of the VBD (FKB-20) including the two reference samples and Student-*t* test including the Cut-off values.

TP and Reference Samples	n	M (SD)	Mdn	Min	Max	*p*-Value (TP1, TP2, TP3)
RBI						
TP1	82	24.1 (9.0)	22	10	48	
TP2	80	22.8 (8.2)	21	10	49	
TP3	55	22.0 (7.7)	20	12	44	
^1^ medical students	56	20.6 (7.1)	18	11	42	<0.001 *, <0.001 *, <0.001 *
^2^ patients	253	27.1 (8.7)	26	12	50	0.013 *, <0.001 *, <0.001 *
VBD						
TP1	82	33.2 (6.6)	34	14	47	
TP2	78	33.7 (6.6)	34	18	50	
TP3	55	33.4 (6.5)	33	16	45	
^1^ medical students	304	36.6 (5.8)	37	19	48	<0.001 *, <0.001 *, <0.001 *
^2^ patients	224	27.0 (7.4)	27	19	48	<0.001 *, <0.001 *, <0.001 *
Cut off for body image disturbance: <31						0.001 *, <0.001 *, 0.004 *

Notes: VBD = vital body dynamics, Student-*t* test. * *p*-value < 0.05. ^1,2^ reference sample: Clement et al. [23]. notes: RBI = rejecting body evaluation. Wilcoxon signed-rank test. * *p*-value < 0.05. ^1,2^ reference sample: Clement et al. [23].

**Table 9 jcm-10-01269-t009:** Descriptive analysis of the individual scores (FSFI-d) compared to the reference sample and Wilcoxon signed-rank test of the FSFI-d domain scores and descriptive analysis of the total scores in comparison with the reference sample (FSFI-d) and Wilcoxon signed-rank test of the FSFI-d total scores.

Domains, Single Scores	*n*	M (SD)	Mdn	Min	Max	*p*-Value
desire						
TP1	76	3.3 (1.2)	3.6	1.2	6	<0.001 *
TP2	76	3.7 (1.2)	3.6	1.2	6	0.034 *
TP3	55	3.7 (1.2)	3.6	1.2	6	0.01
reference sample ^1^	131	4.1 (1.1)				
arousal						
TP1	68	3.0 (2.2)	3.8	0	6	<0.001 *
TP2	69	4.0 (2.1)	5.1	0	6	0.036 *
TP3	52	4.1 (2.0)	4.8	0	6	0.018 *
reference sample ^1^	130	5.0 (1.1)				
lubrication						
TP1	70	3.2 (2.6)	4.5	0	6	<0.001 *
TP2	70	4.6 (2.3)	5.7	0	6	0.095
TP3	51	4.5 (2.3)	5.7	0	6	0.012 *
reference sample ^1^	130	5.6 (1.0)				
orgasm						
TP1	66	2.4 (2.2)	2.8	0	6	<0.001 *
TP2	67	3.9 (2.1)	4.8	0	6	<0.001 *
TP3	51	3.7 (2.2)	4.8	0	6	<0.001 *
reference sample ^1^	129	5.1 (1.3)				
global satisfaction						
TP1	50	3.9 (1.7)	4	0.8	6	1
TP2	56	5.2 (1.1)	5.6	1.6	6	0.019 *
TP3	39	5.2 (1.0)	5.6	1.6	6	0.128
reference sample ^1^	130	5.1 (1.2)				
pain						
TP1	66	0.8 (1.5)	0	0	6	<0.001 *
TP2	69	2.4 (2.1)	2	0	6	<0.001 *
TP3	49	3.7 (2.1)	4	0	6	1
reference sample ^1^	130	5.7 (1.1)				
FSFI-d-Total Score	n	M (SD)	Mdn	Min.	Max.	*p*-Value
TP1	41	19.0 (8.8)	20.8	2	33.6	<0.001 *
TP2	51	27.2 (4.7)	27.9	11.3	35.7	<0.001 *
TP3	38	28.3 (5.3)	28.4	5.2	36	0.007 *
reference sample ^1^	129	30.5 (5.3)				

Notes: Wilcoxon signed-rank test. * *p*-value < 0.05. ^1^ reference sample: Rosen et al. [24].

**Table 10 jcm-10-01269-t010:** Frequency and percentages of the FSFI-d total scores representing grades of sexual dysfunction.

FSFI-d Total Score	TP1	TP2	TP3
Wiegel et al. (106)	*n* = 41	*n* = 51	*n* = 38
<26.55 (sexual dysfunction)	33 (80%)	25 (49%)	13 (34%)
>26.55 (normal sexual function)	8 (20%)	26 (51%)	25 (66%)
Communal et al. (107)			
≤23 (severe sexual dysfunction)	26 (63%)	7 (14%)	4 (10%)
24–29 (moderate sexual dysfunction)	14 (34%)	30 (59%)	19 (50%)
≥30 (normal sexual function)	1 (2%)	14 (27%)	15 (40%)

**Table 11 jcm-10-01269-t011:** Descriptive analysis and Wilcoxon signed-rank test for the PCS and for the MCS of the SF-12 compared to one healthy and one chronically ill reference sample stratified by age.

TP and Reference Samples	n	M (SD)	Mdn	Min	Max	*p*-Value (TP1, TP2, TP3)
**PCS**						
TP1	73	55.9 (5.1)	56.1	37.3	65.4	
TP2	76	54.7 (5.0)	55.9	36.6	62.4	
TP3	49	54.8 (5.8)	55.4	26.1	63.3	
^1^ healthy	123	54.0 (5.9)		35	63.7	<0.001 *, <0.001 *, 0.003 *
^2^ healthy	473	52.9 (6.8)		16.2	62.8	<0.001 *, <0.001 *, <0.001 *
^3^ chronically ill	46	52.4 (7.5)		35	63.7	<0.001 *, <0.001 *, <0.001 *
^4^ chronically ill	227	51.1 (8.6)		16.2	62.8	<0.001 *, <0.001 *, <0.001 *
**MCS**						
TP1	73	42.3 (11.8)	45.6	15.9	57.8	
TP2	76	46.7 (10.3)	49.8	12.2	59.9	
TP3	49	46.7 (10.9)	51.2	22	57.9	
^1^ healthy	123	52.6 (9.1)		12.6	62	<0.001 *, <0.001 *, 0.005 *
^2^ healthy	473	51.9 (8.4)		16.4	71.8	<0.001 *, <0.001 *, 0.023 *
^3^ chronically ill	46	49.4 (11.1)		12.6	62	<0.001 *, 0.230, 0.327
^4^ chronically ill	227	50.3 (9.5)		16.4	71.8	<0.001 *, 0.031 *, 0.158

Notes: Wilcoxon signed-rank test. * *p*-value < 0.05.^1,2^ reference sample healthy: 14–20 years of age, 21–30 years of age, Bullinger et al. [26]. ^3,4^ reference sample chronically ill: 14–20 years of age, 21–30 years of age, Bullinger et al. [26].

**Table 12 jcm-10-01269-t012:** Descriptive analysis and Wilcoxon signed-rank test of the SESA total score compared to the healthy and depressive reference samples.

TP and Reference Samples	*n*	M (SD)	Mdn	Min	Max	*p*-Value (TP1, TP2, TP3)
SESA total score						
TP1	73	107.0 (20.6)	110	47	141	
TP2	65	107.1 (23.5)	111	50	144	
TP3	52	108.0 (22.4)	111	43	142	
^1^ healthy	311	112.3 (15.9)				0.045 *, 0.125, 0.173
^2^ depressive	45	86.1 (18.8)				<0.001 *, <0.001 *, <0.001 *

Notes: Wilcoxon signed-rank test. * *p*-value < 0.05. ^1,2^ reference sample: Sorembe et al. [27].

**Table 13 jcm-10-01269-t013:** Frequencies of the indicators of the SESA total score.

Grade of Self-Acceptance	TP1	TP2	TP3
	*n* = 73	*n* = 65	*n* = 52
Normal self-acceptance	51 (70%)	46 (71%)	36 (69%)
Moderate self-acceptance	15 (20%)	7 (11%)	9 (17%)
Indicator for low self-acceptance	7 (10%)	12 (18%)	7 (14%)

**Table 14 jcm-10-01269-t014:** SESA indicators over the time.

Pat. No	TP1	TP2	TP3
2	102	97	73
3	87	89	93
5	83	84	73
6	*	73	*
8	87	72	129
9	83	71	43
11	87	106	91
12	120	124	96
13	100	115	94
15	94	73	100
17	111	131	82
18	76	76	85
20	101	107	89
23	82	114	106
28	*	*	72
29	88	*	96
34	84	99	99
36	76	63	85
38	72	78	71
43	59	59	73
48	73	94	97
54	93	70	73
59	82	108	**
62	104	88	**
68	107	86	**
70	85	106	**
71	75	80	**
73	111	57	**
76	95	110	**
77	92	*	**
78	91	96	**
79	47	50	**
81	108	91	**

Notes: White represents normal self-acceptance, light grey represents moderate self-acceptance and grey means indicator for low welf-acceptance. * questionnaire excluded, because more than two answers are missing. ** no completed questionnaire.

## Data Availability

The data presented in this study are available on request from the corresponding author.

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
