# Peer review of "Long Term Findings Concerning the Mental and Physical Condition, Quality of Life and Sexuality after Laparoscopically Assisted Creation of a Neovagina (Modified Vecchietti Technique) in Young MRKHS (Mayer-Rokitansky-Küster-Hauser-Syndrome) Patients"

_jcm, 2021, doi:10.3390/jcm10061269_

Round 1

Reviewer 1 Report

This is a great paper on a wide variety of aspects after surgery for MRKH syndrome. Congratulations to the authors. 

I find the results ( tables) a bit hard to read, one after one, maybe it would be possible to combine some of the results according to the psychological constructs.

(combine at least data from one questionnaire into one table, and not where it appears in the text, i.e. FSFI. Also a total reduction of tables would be easier to read, i.e. to combine some of the informations out of tables 1-8)

Author Response

Response to Reviewer 1 Comments

Point 1: I find the results ( tables) a bit hard to read, one after one, maybe it would be possible to combine some of the results according to the psychological constructs. (combine at least data from one questionnaire into one table, and not where it appears in the text, i.e. FSFI. Also a total reduction of tables would be easier to read, i.e. to combine some of the informations out of tables 1-8)

 Response 1: Thank you very much to reviewer 1 for the useful remarks.

  • For the PHQ-D subheadings were partly removed in order to have a continuous flow in the text. The data for somatic symptoms and depression of the PHQ-D were combined within 1 table (now table 6).
  • For the FKB-20 subheadings were partly removed in order to have a continuous flow in the text.The data of the descriptive evaluation of the RBI and VBD scale (FKB-20) was combined in 1 table 8.
  • For the FSFI-d subheadings were partly removed in order to have a continuous flow in the text. Table 9 summarizes FSFI-d single and total scores.
  • For the SF-12 subheadings were partly removed in order to have a continuous flow in the text. Data of the descriptive analysis and Wilcoxon signed-rank test for the PCS and for the MCS oft he SF-12 were combined in table 11.
  • For the SESA subheadings were partly removed in order to have a continuous flow in the text.
  • Tables 2, 4 and 6 were removed and the information included in the text.

Reviewer 2 Report

Dear Authors,

Congratulations for your work. I find it really high quality and valuable. I would like to present you my minor remarks, hoping they will improve the final version of the manuscript:

  1. Figures 2, 3 and 4 seem to be a bit overloaded. I understand that the distribution of data depends on specific scales total scores, however, the distribution and cut-offs do not present very clear in the draft.
  2. I do not fully understand the importance of presenting SESA indicators over time presented in Table 21. The interpretation of these data is missing.
  3. In the paragraph 3.6.1 Characteristics of the SESA reference samples there is wrong table number (44 instead of 19).

Best regards,

Author Response

Response to Reviewer 2 Comments

Point 1: Figures 2, 3 and 4 seem to be a bit overloaded. I understand that the distribution of data depends on specific scales total scores, however, the distribution and cut-offs do not present very clear in the draft.

Response 1: Thank you, we appreciate the useful comments of reviewer 2.

Although we thought about it, we did not find a better type of figure to show the data more clearly due to the large amount of data. The figures are just meant to support the data given in the tables. Alternatively, the figures can be removed or moved into supplementary data.

Point 2: I do not fully understand the importance of presenting SESA indicators over time presented in Table 21. The interpretation of these data is missing.

Response 2: Additionally, the MRKHS patients were individually evaluated at all three timepoints for indicators for low self-acceptance. The description and interpretation is given in the paragraph preceding the table.

Point 3: In the paragraph 3.6.1 Characteristics of the SESA reference samples there is wrong table number (44 instead of 19).

Response 3: The tables have been reduced and renumbered and the correct number is given.

Reviewer 3 Report

This is an interesting study showing the mental and physical condition, quality of life and sexuality after laparoscopically assisted creation of a neovagina in the high number of young patients with a rare disease – MRKHS.

The manuscript is well written and thoroughly prepared. The results are very interesting, still there should be done some minor corrections.

  • Neocope – please explain this term, when it appears for the first time.
  • Experimental section – I propose to change the name of this section to “Study group and methods”, because no experiment is presented here.
  • There is no description of medical examination of the patients – if possible please add a short information.
  • There is no information on co-morbidities – please add if possible.

Author Response

Response to Reviewer 3 Comments

Point 1: Neocope – please explain this term, when it appears for the first time.

Response 1: We thank reviewer 3 for the useful hints.

An additional explaination of the name is now given under measures.

Point 2: Experimental section – I propose to change the name of this section to “Study group and methods”, because no experiment is presented here.

Response 2: The name of the section has been changed accordingly.

Point 3: There is no description of medical examination of the patients – if possible please add a short information.

Response 3: A short information about the gynecological examination was added under 2.1 design an participants.

Point 4: There is no information on co-morbidities – please add if possible.

Response 4: Apart from the associated malformations given under 3.2.2 there were no known relevant comorbities, especially no mental/ psychiatric illnesses.